# JourneyBench: A Challenging One-Stop Vision-Language Understanding Benchmark of Generated Images

Zhecan Wang♠ *    Junzhang Liu♠ *    Chia-Wei Tang†    Hani Alomari†

Anushka Sivakumar†    Rui Sun♠    Wenhao Li♠    Md. Atabuzzaman†    Hammad Ayyubi♠

Haoxuan You♠    Alvi Ishmam†    Kai-Wei Chang♦    Shih-Fu Chang♠    Chris Thomas†

♠Columbia University    ♦UCLA    †Virginia Tech
https://journeybench.github.io/

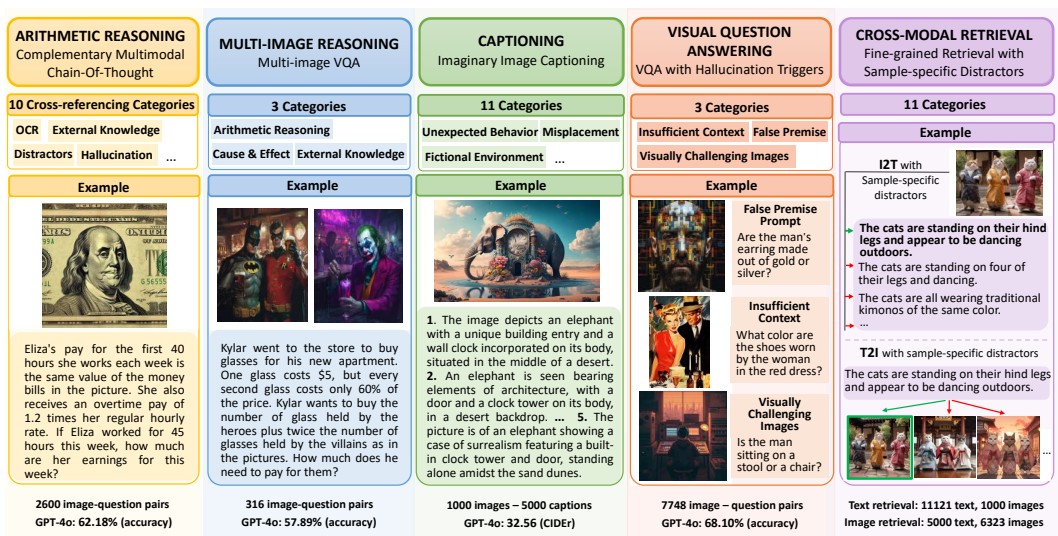

Figure 1: **JourneyBench Tasks with Fine-grained Categories and Example Data**. JourneyBench includes five fundamental vision-language understanding tasks with unconventional imaginary images to test the limits of models' biases, hallucination tendencies, and fine-grained perception abilities.

## Abstract

Existing vision-language understanding benchmarks largely consist of images of objects in their usual contexts. As a consequence, recent multimodal large language models can perform well with only a shallow visual understanding by relying on background language biases. Thus, strong performance on these benchmarks does not necessarily correlate with strong visual understanding. In this paper, we release JourneyBench, a comprehensive human-annotated benchmark of generated images designed to assess the model's fine-grained multimodal reasoning abilities across five tasks: complementary multimodal chain of thought, multi-image VQA, imaginary image captioning, VQA with hallucination triggers, and fine-grained retrieval with sample-specific distractors. Unlike existing benchmarks, JourneyBench explicitly requires fine-grained multimodal reasoning in unusual imaginary scenarios where language bias and holistic image gist are insufficient. We benchmark state-of-the-art models on JourneyBench and analyze performance along a number of fine-grained dimensions. Results across all five tasks show that JourneyBench is

*Equal contribution. Correspondence to: zw2627@columbia.edu

38th Conference on Neural Information Processing Systems (NeurIPS 2024) Track on Datasets and Benchmarks.

exceptionally challenging for even the best models, indicating that models' visual reasoning abilities are not as strong as they first appear. We discuss the implications of our findings and propose avenues for further research.

# 1 Introduction

Multimodal large language models (MLLMs) combine the reasoning capabilities of LLMs with visual (and/or other) modalities, enabling them to tackle a wide array of tasks requiring multimodal understanding, such as visual question answering (VQA) (20; 61; 23), multimodal chain-of-thought reasoning (8; 64), text-to-image generation (59; 42) image captioning (10; 58), and so on. Their impressive performance has led to rapid adoption in our daily life for various tasks such as mathematical reasoning (35; 36), navigation (65; 9), and robotic control (18; 16). This necessitates their rigorous evaluation before deployment in production systems.

While existing Visual Language Understanding (VLU) benchmarks (60; 19; 34) have driven significant progress, they mostly contain limited visual diversity and less complex scenarios than encountered in daily life. For example, many benchmarks restrict their image distribution to resources like COCO (10) or Flickr (58) due to copyright constraints on internet-harvested images. As a result, these benchmarks tend to emphasize commonly occurring subjects, predicates, and objects, over unusual or abstract scenes. This enables models to excel by leveraging previously acquired common-world knowledge without necessarily understanding the actual content of the images. While this bias might inflate scores on academic benchmarks, it can lead to significant challenges when transitioning to real-world applications (43). Moreover, benchmarks curated to evaluate Multimodal Chain-of-Thought (MCOT) reasoning such as (36), often feature redundant visual content (i.e. not needed to answer the question), as illustrated in Figure 3. Current MCOT benchmarks also fail to adequately address critical issues like hallucination (32) and prediction consistency. On retrieval benchmarks, models' performance is saturating near human-level (10; 58), making it challenging to distinguish between models. This saturation is partly due to the lack of fine-grained detail in current retrieval benchmarks, which do not sufficiently challenge today's powerful models (47).

The rise of prompt-based generated images presents a unique opportunity for a comprehensive multimodal benchmark. Unlike real images, these generated images bypass copyright issues and offer diverse visual content, enabling more challenging and nuanced testing scenarios. Generated images can combine uncommon concepts, such as "elephant on macaroons" which are rare in traditional datasets but critical for evaluating a model's true understanding of visual concepts. For example, COCO contains object relations found in ConceptNet (33) 68% of the time vs. only 6% in the generated images we collect. Further, as generated images become increasingly realistic and proliferate online, incorporating them into benchmarks for assessing models' capabilities to understand and interpret diverse visual scenes will become increasingly important. By leveraging prompt-based generated images, we can address the limitations of existing benchmarks, providing better controllability and diversity in visual content. This approach enables rigorous testing of models' hallucination tendencies, consistency, and ability to function effectively in varied and unpredictable environments.

With this insight, we present **JourneyBench**, a comprehensive VLU benchmark leveraging prompt-based generated images within a novel human-machine-in-the-loop (HMIL) framework. While some recent works leveraging generated images have been proposed, they are either on a small scale (6) (e.g.~1K samples) or not challenging and comprehensive enough (40). In contrast, JourneyBench is large (~13.5K samples) and evaluates models' advanced reasoning capabilities across five challenging tasks: MCOT, multi-image MCOT (MMCOT), fine-grained cross-modal retrieval (CR), open-ended visual question answering (VQA) with hallucination triggers[2], and imaginary image captioning. It specifically assesses models' hallucination tendencies, prediction consistency, and ability to understand and differentiate fine-grained details. Our contributions are as follows:

- We introduce JourneyBench, a comprehensive, expertly annotated, challenging VLU benchmark of imaginary images to rigorously test models' capabilities across five tasks.

---

[2]Similar to other recent benchmarks (35), JourneyBench builds on top of a prior, unpublished benchmark (by the authors) for VQA with hallucination triggers called HaloQuest. We include a complete write-up in our supp. and do not repeat details here. All other components of JourneyBench are new and described herein.

- To the best of our knowledge, for the first time, we address VLU evaluation with imaginary (unusual or fictional) images on a large scale. We further contribute the challenging complementary MCOT, nvoel multi-image MCOT and fine-grained retrieval tasks with generated images.
- We develop a novel adversarial HMIL framework to scale up the generation of high-quality data.
- We conduct detailed analyses to provide insights into model performance, behavior and limitations. For instance, even the powerful model GPT-4, achieves only 57.89% accuracy on multi-image VQA and struggles with co-referencing across modalities in MCOT, achieving just 62.18% accuracy.

## 2 Related Works

VLU evaluation has been a crucial tool in assessing AI performance across various tasks(63), including cross-modal retrieval (10; 58; 14), MCOT (36; 8; 35; 64; 60), image captioning (10; 58), visual question answering (VQA) (20; 23; 61; 38; 45; 5), and multi-image visual reasoning (56; 25; 51). Despite their significance, there have been limited efforts (6; 40) to leverage generated images in VLU evaluation. These attempts have not fully exploited the controllability, convenience, and strengths of prompt-based generated images (44; 4) to address more challenging issues such as MCOT, fine-grained cross-modal retrieval (48; 68), and multi-image visual reasoning (56; 25; 51). Cross-modal retrieval is a fundamental capability of AI with applications in many domains (67). However, recent models' performances have plateaued on existing benchmarks (10; 58; 14), which primarily focus on differentiating non-related image-text pairs. This allows models to succeed by memorizing holistic styles or content without paying attention to fine-grained visual details (48; 68). Our fine-grained multimodal retrieval task, on the other hand, uses prompt-based generated images to create sample-specific distractors, challenging models to differentiate intricate details. MCOT is another challenging task that involves reasoning across visual and textual modalities. Existing VQA and MCOT datasets often include redundant images, allowing models to solve problems using text inputs alone (53; 36; 35). Furthermore, these datasets fail to address hallucination and consistency issues in real-world math problems (21; 37; 24; 22; 64). To tackle these limitations, we develop complementary MCOT questions that require the integration of information from both modalities. Additionally, by pairing the same math reasoning question with different visual contexts, we can assess models' consistency and behavior, leveraging the flexibility of generated images. While many existing datasets for image captioning (10; 58; 14) and VQA (20; 23; 61; 38; 45; 5) focus on everyday scenarios with real images, our tasks—imaginary image captioning and HaloQuest (52) —aim to evaluate models' understanding of imaginary images, including unusual and fictional visual scenes. By harnessing the strengths of prompt-based generated images, we enhance these popular VLU tasks to push the boundaries of benchmarking high-performing models.

## 3 JourneyBench

In this section, we discuss the procedure for constructing JourneyBench. We first describe our approach to collecting high-quality, diverse, and interesting images. Then, we detail the annotation process for each of the five tasks. We include further details of our dataset, like quality assurance via multiple rounds of annotations, consistency checks, and dataset statistics, in the appendix. Collectively, JourneyBench's curation involved over *2,200 hours* of human annotation effort.

### 3.1 Data acquisition and filtering

***Retrieving generated images.*** We aim to create a VLU benchmark containing challenging and diverse imaginary images, including unusual, abstract, and complex ones by leveraging the advantages of prompt-based generated images. However, generated images tend to suffer from low quality and biased distribution. To prevent that, we instead *retrieve* popular prompt-based generated images from Midjourney (4) - a large crowd-based platform - using web scraping tools with metadata information. We ensure the diversity of image content by adopting the strategy from (52) – combining 17 topic words and 15 attribute words to form the query used to retrieve images. This approach results in a significantly larger and more diverse set of topic words for image content compared to previous image-text datasets [3].

---

[3]Detaiued analysis in Section 4.4 and appendix.

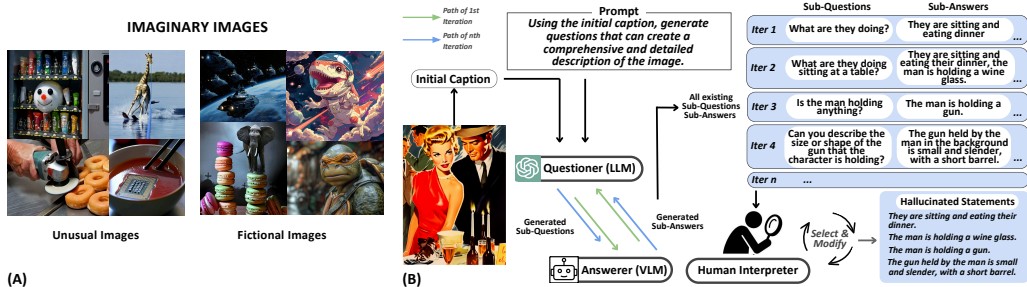

Figure 2: **Examples of Imaginary Images** and **Human-Machine-in-the-Loop Pipeline.**

*Image filtering.* Human annotators select images from the retrieved pool that are: **unusual**, **fictional** (unrealistic), and contain visually **comprehensible** concepts. Unusual images depict scenarios outside of everyday experiences, feature unexpected juxtapositions of objects, or include visually striking elements. Fictional images present unrealistic or impossible scenes (*e.g.*, an elephant standing on macaroons). Comprehensibility ensures that images are free of artifacts and understandable to humans. This balances the fine dynamics between creating challenging scenarios and ensuring legible visual concepts to reliably test models. We present annotators with a set of questions to help them identify if images fulfill these three criteria. To address human subjectivity in this task , we employ at least four Amazon Mechanical Turk (MTurk) annotators for each image. They achieve 100% agreement in over 72% of cases. Detailed information about the user interface, data filtering process, and questions are provided in the appendix.

*Categories of imaginary images* Providing a fine-grained categorization of imaginary images can assist in our understanding of models' behaviors across categories of unfamiliar scenarios. Hence, we categorize our images based on how unusual or how unrealistic they are. Because of the subjective nature of this problem, we hire four experienced co-author annotators who collectively converged on 15 categories of unusualness and unrealisticness across images, as listed in the axes of Figure 4, which were then used to annotate the dataset.

We next present how we use imaginary images to form challenging VLU tasks within JourneyBench.

## 3.2 Imaginary Image Captioning

While captioning is a standard task for VLU benchmarking, we seek to test models' abilities to understand and caption *imaginary* images in JourneyBench. To this end, we require models to generate a single-sentence description of an image highlighting elements that make it imaginary. The ground truth annotation of each collected imaginary image is written by eight MTurk annotators to describe the most unusual or fictional part of the image. Then the captions are verified by another four experienced MTurk annotators to avoid subjective biases among annotators. The user interfaces and detailed procedures during the annotation process are in the appendix.

## 3.3 Fine-grained Cross-modal Retrieval

Cross-modal retrieval is a fundamental VLU task included in many benchmarks (10; 58; 14). Given an image, the objective is to retrieve the matching text, and vice versa. This capability is critical for AI models in various domains, including search engines. However, the performance of existing models on popular cross-modal retrieval benchmarks such as MS-COCO (10) and Flickr30K (58) has reached saturation (27). These benchmarks primarily involve real images and focus on largely discriminating between pairs holistically. For example, in image-to-text retrieval, other images' matching texts are treated as distractors (i.e. negatives), even though they are largely irrelevant to the target image, making the task easier. However, for models to accurately retrieve relevant content, it is crucial to be able to differentiate image-text pairs at a fine-grained level. Thus, to challenge models' ability to perform fine-grained differentiation across similar images, we propose an adversarial HMIL framework to create sample-specific distractors, i.e. hard negatives which require fine-grained discrimination to overcome, for each query sample. For instance, as illustrated in the rightmost examples in Figure 1, our framework creates challenging scenarios requiring models to focus on intricate details to successfully retrieve the correct image-text pairs. We next describe our data curation and annotation approach below for each retrieval direction.

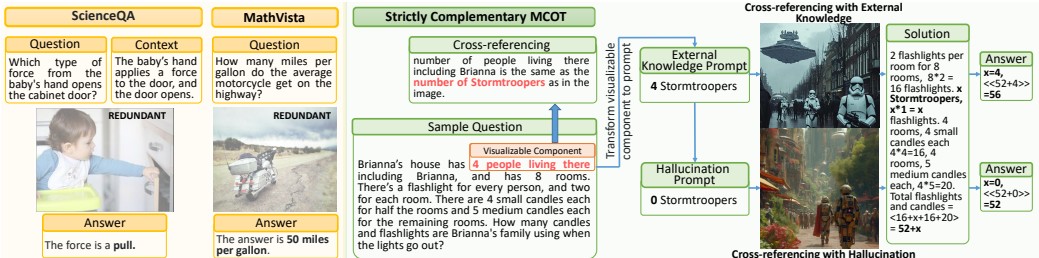

Figure 3: **Comparison between ScienceQA, MathVista (left), and our Strictly Complementary MCOT (right) with Examples.** While ScienceQA and MathVista images provide redundant visual information, Journeybench provides complimentary visual information that is necessary to answer the question. This ensures a more rigorous evaluation of multimodal reasoning capabilities.

*Image-to-Text retrieval.* We experiment with two HMIL approaches to scale up and generate distractors. In the first one, we feed the ground-truth caption (Sec.3.2) into MLLMs like GPT-4V and prompt them to generate relevant but conflicting hallucinated statements using in-context examples. Human annotators then verify these generated distractors. This approach is effective but has limitations. It performs well when the image is easily comprehensible by the MLLMs and the ground-truth caption is detailed. However, the generated distractors are often not challenging enough and somewhat obvious, as the conflicting elements are "guessed" by the generation model, which itself introduces bias. We find in cases where the image is complex, or the ground-truth text is not detailed, the model often introduces irrelevant elements into the distractors, reducing their quality.

To address these limitations, we develop a more effective HMIL system inspired by (57) that introduces a dialogue between an LLM and an MLLM. As in Figure 2, the process begins by feeding the initial ground-truth caption and the prompt into the LLM, which generates questions about the image that are answered by the MLLM. With each iteration, the MLLM-LLM's errors propagate, making the hallucinated predictions more difficult to overturn and thus revealing "blind spots" to humans. These "blind spots" are not merely imagined by the generators but empirically demonstrated on the task. Human annotators then pinpoint these spots, collecting hallucinated answers or statements as potential distractors. We found this HMIL approach generates high-quality distractors with relevant but conflicting details that are challenging for models to notice.

*Text-to-Image retrieval.* Similar to image-to-text retrieval, for each target text, we use the matching ground-truth image to obtain sample-specific image distractors. We employ a group of expert annotators to query the Midjourney platform to retrieve relevant but conflicting image distractors for each sample. During this process, annotators are asked to find image distractors based on two criteria: the subject, the composition, or both. For example, as illustrated in the bottom rightmost image in Figure 1, for the subject criterion, annotators should find image distractors that also feature three cats. For the composition criterion, they should find image distractors where there are three animals positioned side by side and facing the camera. By adhering to these criteria, we ensure that annotators collect high-quality image distractors that cannot be easily differentiated without fine-grained details. On average, for each target text, we obtain about five sample-specific distractors.

### 3.4 Complementary Multimodal Chain-of-Thought

In the MCOT task, the input consists of an image and a question which requires the model to integrate information from both modalities. However, existing MCOT resources like MathVista (35) and ScienceQA (36) often contain redundant visual information, allowing models to answer questions using only the language input. To address this, we aim to build a **Strictly Complementary** MCOT dataset that *requires* multimodal reasoning. In this dataset, visual and text information will be complementary, ensuring models must co-reference both modalities for chain-of-thought reasoning. Our experiments reveal that multimodal co-referencing during the chain-of-thought process is very challenging for existing models. For example, GPT-4 achieves over 90% accuracy on the text-only version of our COT questions, GSM8K (15), but only 49.34% and 61.2% in our strictly complementary MCOT setting for GPT-4V and GPT-4o, respectively. This significant drop highlights the importance of our complementary MCOT dataset in evaluating multimodal reasoning capabilities.

***Visualizing text-only MCOT.*** We scale up the generation of strictly complementary MCOT data by converting the text-only COT benchmark, GSM8K (15), into MCOT using prompt-based generated images. As shown in Figure 3, the process begins by identifying visualizable text components and converting them into prompts to generate images. These images replace the identified text components with new text requiring co-referencing the image. This method rapidly scales up the creation of high-quality, complementary MCOT data which allows testing of models' multimodal reasoning capabilities in solving arithmetic problems.

***Co-referencing categories.*** Generated images' controllability allows us to test each question with diverse visual contexts all requiring the same arithmetic reasoning logic to better understand models' abilities. As shown in Figure 5, we evaluate models' ability to co-reference visual content requiring external knowledge for arithmetic problems and assess hallucination tendencies by omitting referenced objects. Despite recent MLLM progress in MCOT benchmarks, co-referencing remains extremely challenging. We categorize types of co-referencing to analyze models' weaknesses in Figure 5. Our appendix contains detailed definitions of each type shown. Our findings indicate models struggle with hallucination and using external knowledge in the MCOT task, highlighting the need for further research.

### 3.5 Multi-image Visual Question Answering

Recently, benchmarks for multi-image VQA have been proposed (25; 46), requiring models to reason over multiple images for VQA. However, due to limited real image resources, existing datasets primarily test basic abilities like color matching, image-text matching, and object counting. In contrast, our multi-image VQA task evaluates three specific and challenging reasoning categories: arithmetic reasoning, applying external knowledge to visual reasoning, and identifying cause and effect, as shown in the example of Figure 1.

For multi-image VQA data requiring arithmetic reasoning, we use a similar approach to our single-image MCOT data collection. For data requiring external knowledge, we engage six expert annotators to identify and collect high-quality Midjourney images that require external knowledge to understand. These annotators then generate multi-image visual questions based on these images. For the cause-and-effect category, we use prompt-based generated images to convert the text-only cause-and-effect dataset, COPA (7). Each COPA sample contains two text events representing cause and effect. Annotators identify samples with visualizable events and obtain corresponding generated images, which are then compiled into multi-image samples to test if models can identify the cause or effect between visual events. Our multi-image VQA setting challenges even the best models with complex reasoning tasks requiring co-referencing, applying external knowledge, and understanding cause-and-effect relationships across multiple images.

## 4 Experiment

### 4.1 Evaluation Metrics

For cross-modal retrieval, we report Recall@k (R@k) for $k \in 1, 5, 10$. For captioning, we report the standard BLEU, ROUGE, CIDEr, and Meteor scores. For our MCOT and multi-image VQA tasks, we use Llama-3-8B (3) to extract the answers from the models' generated solutions and then again ask Llama-3-8B to determine if the answer is correct by providing the question and ground truth answer with the prompt. We then use Llama-3-70B for solution verification by asking Llama to verify if the generated solution follows the logic of the ground truth solution. We manually verified a subset of Llama-3's responses to ensure quality. In the appendix, we provide additional details of our evaluation setup, along with the prompts used.

### 4.2 Baseline Models

For our retrieval tasks, we employ SOTA retrieval pre-trained models, including ALBEF (30), CLIP (41), $X^2$-VLM (Large) (62), BEiT3 (50), BLIP2 (29), OpenCLIP-Coca (13), and InternVL (12). In the case of MCOT, multi-image VQA, and captioning tasks, we leverage current SOTA vision-language generative models in a zero-shot manner, along with GPT-4o (1) and GPT-4V (2) . The models utilized for these tasks include LLaVA-NeXT (28), VILA (31), BLIP-2 (29), Mantis (25), InternVL (11), MiniGPT-4 (66), mPLUG-Owl (54), mPlug-Owl2 (55), Idefics2 (26), and CogVLM2

(49). We use different versions and sizes of these models with our fixed prompts, and the details can be found in the appendix.

| Model | Text Retrieval | | | | | | | | Image Retrieval | | | | | | | |
|---|---|---|---|---|---|---|---|---|---|---|---|---|---|---|---|---|
| | Flickr30K(1K) | | MS-COCO(1K) | | JourneyBench(1K) w/o distractors | | JourneyBench(1K) w/ distractors | | Flickr30K(1K) | | MS-COCO(1K) | | JourneyBench(1K) w/o distractors | | JourneyBench(1K) w/ distractors | |
| | R@1 | R@5 | R@1 | R@5 | R@1 | R@5 | R@1 | R@5 | R@1 | R@5 | R@1 | R@5 | R@1 | R@5 | R@1 | R@5 |
| ALBEF-210M (30) | 88.50 | 98.50 | 89.10 | 98.30 | 72.30 | 86.10 | 65.36 | 83.75 | 75.90 | 92.60 | 72.28 | 94.18 | 66.12 | 88.65 | 50.02 | 75.46 |
| CLIP-430M (41) | 85.30 | 97.90 | 75.60 | 93.20 | 70.60 | 85.70 | 60.80 | 83.30 | 64.90 | 87.20 | 54.50 | 81.80 | 66.80 | 88.80 | 51.20 | 76.50 |
| $X^2$-VLM-Large-590M (62) | **98.80** | **100.00** | **93.60** | **99.50** | 78.54 | 92.78 | 64.97 | **90.47** | **91.80** | **98.60** | **83.32** | **96.86** | 75.04 | 93.16 | 61.02 | **85.00** |
| BEiT3-674M (50) | 89.50 | 98.80 | 81.10 | 96.60 | 74.10 | **87.80** | 65.90 | 86.10 | 75.94 | 93.34 | 66.40 | 89.50 | 68.00 | 90.30 | 56.20 | 79.90 |
| BLIP2-12B (29) | 92.80 | 99.90 | 91.30 | 99.10 | **81.29** | 95.17 | 63.78 | 87.76 | 89.70 | 98.10 | 78.78 | 94.92 | 75.77 | 91.66 | 59.97 | 82.48 |
| OpenCLIP-CoCa-13B (13) | 92.50 | 99.50 | 75.89 | 93.63 | 70.43 | 85.41 | 60.04 | 83.32 | 80.40 | 95.70 | 59.30 | 85.51 | 65.83 | 86.66 | 48.70 | 72.56 |
| InternVL-C-13B (12) | 94.70 | 99.60 | 85.34 | 96.86 | 78.22 | 89.21 | **67.73** | 86.41 | 81.70 | 96.00 | 71.43 | 91.50 | 75.84 | 93.34 | 62.29 | 83.44 |
| InternVL-G-14B (12) | 95.70 | 99.70 | 87.58 | 97.64 | 78.52 | 89.81 | 67.53 | 86.51 | 85.00 | 97.00 | 75.64 | 93.77 | **76.80** | **93.80** | **63.71** | 84.84 |

Table 1: **Zero-shot Evaluation of Cross-modal Retrieval.** The best and second-best results are bolded and underlined. The performance of baseline models on JourneyBench without distractors is comparable to that of existing cross-modal retrieval tasks of similar scale, indicating their generalizability to generated images. However, there is a notable decline in performance when distractors are added, highlighting the critical role of sample-specific distractors in enhancing the challenge of the tasks. Additional results available in appendix.

## 4.3 Quantitative Analysis

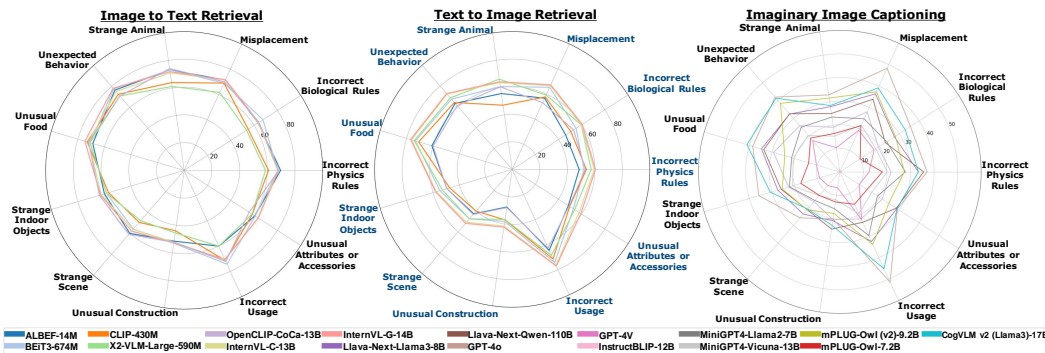

Figure 4: **Zero-shot Evaluation on Fine-grained Categories of Retrieval and Captioning.** I2T (left), T2I (center), and Imaginary Image Captioning (right) are measured by Recall@1, Recall@1, and CIDEr respectively. Models particularly struggle with "Unusual Construction" subcategory.

We experimented with various SOTA models on our newly introduced JourneyBench datasets with a range of different experiments, including cross-modal fine-grained retrieval, imaginary image captioning, and multimodal chain-of-thought and multi-image VQA.

**Models struggle with differentiating fine-grained visual details.** We selected a diverse set of models that have previously exhibited strong performance on established cross-modal retrieval datasets (58; 10; 14). Table 1 presents the results of existing SOTA retrieval models on these datasets and our fine-grained cross-modal retrieval dataset. Among these models, InternVL (12) and BLIP2 (29) achieve the highest R@1 score of 67.63% and 81.29% for text retrieval with and without distractors, respectively. Regarding image retrieval, with and without distractors, InternVL-G-14B (12) achieved the highest R@1 scores. However, as depicted in Figure 4, the performance of these models on our dataset reveals significant challenges and limitations, with the majority of scores clustered around 60% and failing to surpass the 80% mark across all categories.

The lower recall scores in JourneyBench compared to MS-COCO (10) and Flickr30k (58) demonstrate that models encounter greater challenges in retrieving text and images from our dataset. For instance, the highest R@1 performance for text retrieval in MS-COCO-1k is 93.6%, whereas in JourneyBench with and without distractors, it was only 70.1% and 81.29%, respectively. Similarly, for image retrieval, the highest R@1 score on MS-COCO-1k is 83.32%, which is notably higher than the 76.8% and 63.71% scores in our dataset. This disparity highlights the models' struggle in differentiating fine-grained visual and textual details, especially with sample-specific distractors in JourneyBench. The varying performance gaps across categories suggest that certain types of image-text relationships

are more challenging to capture and align, with categories like "Unusual Construction" and "Strange Scene" requiring more sophisticated understanding and reasoning abilities to bridge the semantic gap between the visual and textual modalities.

| Model | BLEU1-4 | CIDEr | METEOR | Rouge |
|---|---|---|---|---|
| MiniGPT4-Lama2-7B (66) | 19.60 | 20.91 | 18.07 | 28.76 |
| mPLUG-Owl-7.2B (54) | 19.53 | 14.68 | 19.32 | 27.66 |
| LLaVA-Next-Llama3-8B (28) | 20.01 | 28.69 | 15.01 | 26.38 |
| mPLUG-Owl (v2)-9.2B (54) | **24.31** | 26.74 | 20.51 | 30.97 |
| Blip-2-12B (29) | 17.75 | 26.00 | **22.00** | **37.00** |
| InstructBLIP-12B (17) | 10.23 | 00.46 | 17.19 | 19.51 |
| OpenCLIP-CoCa-13B (13) | 18.79 | 21.59 | 12.02 | 24.40 |
| MiniGPT4-Vicuna-13B (66) | 12.79 | 16.21 | 17.10 | 24.51 |
| CogVLM v2 (lama3)-17B (49) | 21.86 | 30.31 | 18.63 | 28.67 |
| LLaVA-Next-Qwen110B (28) | 19.73 | 27.18 | 14.96 | 26.61 |
| GPT-4o | 21.86 | **32.56** | 18.56 | 28.37 |
| GPT-4V | 17.36 | 11.24 | 19.47 | 26.75 |

Table 2: Zero-shot Evaluation on Imaginary Image Captioning. The best and second-best results are bolded and underlined. The low scores on the metrics indicate the baselines struggle to describe imaginary images.

**Models are not used to imaginary visual scenarios.** We conducted experiments that included various SOTA models for visual understanding, such as LLaVA-NeXT (28), MiniGPT-4 (66), mPlug-Owl (54; 55), GPT-4o, etc. for the captioning task. In Table 2 and Figure 4, most of the models performed poorly on JourneyBench compared to their performance on other captioning datasets (58; 10; 14), with the majority of the models achieving CIDEr scores less than 30.

**Co-referencing across modalities is challenging in arithmetic reasoning.** Figure 5 illustrates the performance of SOTA methods across fine-grained categories of the JourneyBench MCOT dataset. Our complementary MCOT task proves to be highly challenging, with GPT-4o achieving only 62.18% accuracy. Most other models, except GPTs and LLaVAs, score below 10%. Notably, GPT-4V and GPT-4o struggle with consistency, hallucination, and co-referencing in visual contexts with numerous objects. Additionally, smaller VLMs also find it difficult to utilize external knowledge when solving MCOT questions.

To demonstrate the complementary nature of our image-question pairs in MCOT, we tested a language-only GPT-4o model on our dataset, which resulted in just 16.64% accuracy. In contrast, language-only GPT-4o achieved 83.9% on ScienceQA (36). This significant difference underscores the importance of complementary visual and textual information in multimodal reasoning tasks. The red star in Figure 5 indicates human performance at 84%, suggesting that there is still significant room for improvement even for the SOTA LLMs.

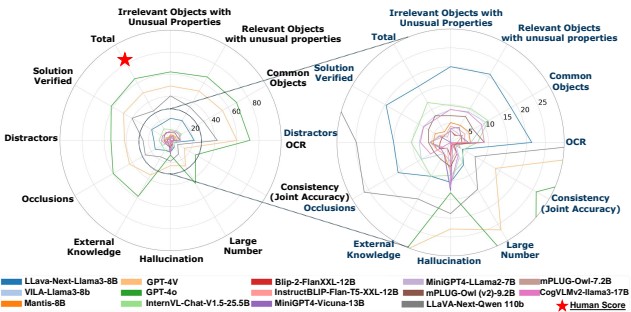

Figure 5: **Zero-shot Evaluation on Fine-grained Categories of MCOT.** Models struggle to get high accuracy in all categories, especially for image-question pairs with hallucinations or with large numbers of objects.

**Co-referencing across multiple images is extremely challenging.** Table 3 presents the performance of different SOTA VLMs on our proposed multi-image VQA dataset across various categories, as well as on the Mantis-Eval dataset. Overall, models encountered greater challenges in co-referencing across multiple images in JourneyBench, with low scores in the range of 39.04% ± 18.85%. Especially concerning MMCOT VQA, performance is even lower in the range of 23.58% ± 19.81% across different

| Model | Multi-Image VQA | | | | | Cause and Effect | Mantis Eval |
|---|---|---|---|---|---|---|---|
| | All | MMCOT | | | | | |
| | | All | Arithmetic Reasoning | External Knowledge | Solution Verification | | |
| VILA-8B (31) | 24.20 | 6.14 | 3.73 | 8.65 | 3.77 | 53.92 | 51.15 |
| Idefics2-8B (26) | 27.82 | 6.61 | 2.81 | 10.57 | 4.95 | 65.03 | 48.85 |
| Mantis-Idefics2-8B (25) | 19.90 | 3.30 | 3.71 | 2.88 | 7.26 | 49.02 | 57.14 |
| Mantis-SigLIP-8B (25) | 23.29 | 4.72 | 5.98 | 3.41 | 7.82 | 55.88 | 59.45 |
| GPT-4V | 48.70 | 32.54 | 32.88 | 32.2 | 36.31 | 77.06 | 62.67 |
| GPT-4o | **56.39** | **41.03** | **52.04** | 29.61 | **43.39** | **83.33** | **73.42** |
| Human | 78.90 | 71.40 | 86.00 | 55.80 | - | 92.00 | - |

Table 3: **Zero-shot Evaluation on Multi-Image Visual Reasoning.** The best and second-best results are bolded and underlined. Models like GPT-4o perform worse on our Multi-image VQA or MMCOT than on Mantis-Eval. Note that most models on Cause and Effect - being a binary-choice question - have an accuracy of nearly random guessing.

SOTA VLMs. Meanwhile, all the models achieved much higher accuracy scores in the range of 61.13% ± 12.29% on the Mantis-Eval dataset. For instance, GPT-4o achieved an accuracy of 73.42% on the Mantis-Eval dataset, which is approximately 32 % and 17% higher than its performance, 41.03% on our MMCOT and 56.39% on our multi-image VQA. Similar to our MCOT task, we also conduct a human evaluation to obtain an estimation of the expected maximum performance. As shown in the figure, the arithmetic reason is similar to MCOT, suggesting humans are indifferent to

multiple images. However, since we restrict access to the internet during the human test, the low external knowledge result causes a significant drawback to the overall score.

### 4.4 Qualitative Analysis

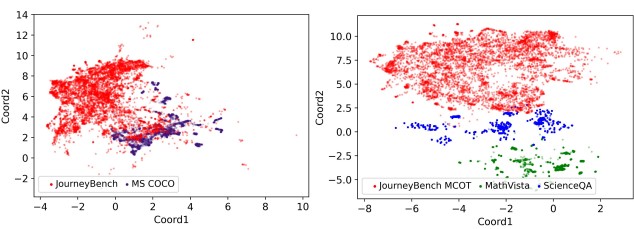

Figure 6: **Low-dimensional Representation of Journey-Bench, MS COCO, MathVista, and ScienceQA Images.** JourneyBench shows a more diverse distribution.

**Image Diversity Visualization.** Figure 6 shows the result of dimension reduction using UMAP (39) on CLIP's embedding space, sampling an equal number of images from each dataset. In the top figure, JourneyBench's distribution is not only more expansive but also encompasses the majority of COCO's data distribution, suggesting a richer semantic diversity. The bottom figure shows JourneyBench's MCOT images have a similarly diverse distribution. Compared to existing MCOT benchmarks like MathVista (35), and ScienceQA (36), JourneyBench MCOT displays significantly greater diversity. Despite sampling an equal number of images from each dataset, JourneyBench appears more populated in the graph. This is because images in MathVista and ScienceQA are often very similar, such as maps, tables, and illustrations that change only slightly, resulting in densely overlapping data points in the UMAP visualization.

## 5 Conclusion

We introduce JourneyBench, a new benchmark that tests models' understanding of unusual or fictional images across various tasks, including multimodal chain-of-thought, multi-image VQA, image captioning, visual question answering, and cross-modal retrieval. JourneyBench's tasks consistently yield lower evaluation scores from all tested baseline models, underscoring the challenges posed by its unusual or fictional image subjects, strategically designed distractors, hallucination-inducing questions, and questions that require cross-modal referencing. This makes JourneyBench an ideal tool for assessing the capabilities of advanced MM-LLMs, pushing the boundaries of what these models can understand and interpret.

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
