# OpenReview forum: "JourneyBench: A Challenging One-Stop Vision-Language Understanding Benchmark of Generated Images"
_NeurIPS.cc/2024/Datasets_and_Benchmarks_Track — NeurIPS 2024 Track Datasets and Benchmarks Poster_

### Official Review · Reviewer_jgCg · 2024-06-22
**A comprehensive benchamrk to evaluate visual language understanding of prompt-based generated images.**

**Rating:** 6
**Confidence:** 5
**Clarity:** The paper is generally well written.

**Review:**

Quality:

The overall quality of this paper is quite good, encompassing a comprehensive benchmark to evaluates models’ advanced reasoning capabilities across five challenging tasks.

Clarity:

The overall presentation of this paper is clear.

Originality:

This paper demonstrates a high level of originality, as it introduces a benchmark of imaginary images to rigorously test models’ capabilities across five tasks.

Significance:

Evaluating fine-grained multimodal reasoning abilities of MLLMs in generated images is significant since they can not rely on common-world knowledge without necessarily understanding the actual content of the images.

**Strengths:**

1. This benchmark covers five challenging tasks including MCOT, multi-image MCOT,  fine-grained cross-modal retrieval, open-ended
visual question answering (VQA) with hallucination triggers, and imaginary image captioning.

2. The proposed human-machine-in-the-loop pipeline is effective to generate high-quality evaluation data.

3. The authors provide detailed analysis of model performance along a number of fine-grained dimensions.

**Additional Feedback:**

Please see Opportunities For Improvement.

**Correctness:**

The evaluation methods and experiment design are appropriate and performed correctly.

**Documentation:**

There is sufficient detail to support reproducibility.

**Ethics:**

There are no ethical concerns with the submission.

**Limitations:**

The authors have discussed the limitations in the paper.

**Opportunities For Improvement:**

1. This paper primarily presents the five evaluation tasks of JourneyBench in a qualitative manner. The authors should provide quantitative data statistics, which would allow us to understand the composition of this benchmark.

2. Is there an inherent connection between the five tasks evaluated by this benchmark? In other words, why did the author choose to evaluate these five tasks? For instance, cross-modal retrieval is not a mainstream task in the MLLM field.

**Relation To Prior Work:**

The difference from previous contributions are clearly discussed.

**Summary And Contributions:**

1. This paper proposes a Visual Language Understanding (VLU) benchmark JourneyBench for imaginary images, covering five tasks and focusing on large-scale evaluation of models with unusual or fictional images.

2. The authors introduce a novel adversarial HMIL framework to scale up the generation of high-quality data.

3. The authors provide detailed analyses of model performance, behavior, and limitations.

---

> ### Author Rebuttal · Authors · 2024-08-19
>
> ## Quantitative Data Statistics and Task Connection
>
> We appreciate the feedback and agree that providing quantitative data statistics is crucial for a thorough understanding of our benchmark. Due to space constraints and the extensive nature of our tasks, we opted to include detailed dataset statistics in the Appendix. We would like to direct the reviewers to Appendix F, where a comprehensive analysis of the dataset, including the distribution of samples across tasks and categories, is provided. However, we recognize the importance of including key statistics in the main text, and we will incorporate a concise summary of the dataset composition and task-specific statistics in the final version of the paper.
>
> Regarding the connection between the five tasks, we carefully selected these tasks to evaluate a broad spectrum of multimodal reasoning capabilities, which are essential in Vision-Language Understanding (VLU). Each task targets a specific dimension of VLU:
> - **Visual Reasoning:** This is evaluated through visual question answering (VQA) tasks with both single and multiple images. These tasks test the models' ability to understand and reason about visual content in different contexts. More specifically, our VQA task evaluates models’ hallucination tendencies.
> - **Quantitative Understanding:** We assess this through chain-of-thought reasoning tasks, where models must apply arithmetic or logical reasoning based on visual inputs. These tasks involve both single and multi-image contexts to evaluate consistency and cross-referencing abilities.
> - **Image-Text Matching:** Fine-grained cross-modal retrieval tasks are included to challenge models in identifying precise relationships between textual descriptions and visual content, pushing beyond surface-level matching to test nuanced understanding.
>
> These tasks collectively cover critical aspects of VLU, including reasoning, comprehension, and retrieval, making the benchmark comprehensive and rigorous. We believe these are the current research thrusts that are broadly being explored in the current literature. We will enhance the introduction to clearly articulate the rationale behind the selection of these tasks, emphasizing their interconnections and their relevance to the broader goals of multimodal reasoning evaluation.

---

> > ### Comment · Reviewer_jgCg · 2024-08-20
> >
> > Thanks for the response. I decided to keep my rating.

---

> > > ### Author Response · Authors · 2024-08-24
> > >
> > > Thank you very much for your time, effort, and valuable insights into our work. Your feedback is greatly appreciated and will help us enhance the quality of our research.

---

### Official Review · Reviewer_njtE · 2024-07-16

**Rating:** 6
**Confidence:** 2
**Clarity:** The paper is well written.

**Review:**

**Strengths**
- The use of generated images allows for diverse and challenging scenarios, free from copyright constraints.
- The inclusion of tasks like VQA with hallucination triggers and fine-grained retrieval addresses critical issues often overlooked in existing benchmarks.
- The paper introduces an HMIL framework to generate high-quality, challenging data, ensuring that the benchmark tasks remain robust and difficult for current models.

**Weaknesses**
- There are no significant issues of this benchmark. It is greatly valuable for exploring the limitations of MLLMs.
- Although the benchmark includes a variety of tasks and scenarios, the total number of samples (~13.5K) might still be insufficient for comprehensive evaluation across all possible edge cases and model capabilities.

**Strengths:**

- JourneyBench fills a gap in current VLU benchmarks by introducing tasks that require fine-grained reasoning with generated images, pushing the boundaries of current model capabilities.
- The benchmark encompasses five different tasks, each targeting specific aspects of multimodal reasoning, from captioning to arithmetic reasoning and cross-modal retrieval.
- The paper provides extensive benchmarks using state-of-the-art models, offering a clear picture of current limitations in multimodal reasoning.

**Additional Feedback:**

N/A

**Correctness:**

The evaluation methods and experiment design appropriate and performed correctly.

**Documentation:**

N/A

**Ethics:**

No ethics concerns.

**Opportunities For Improvement:**

See weaknesses.

**Relation To Prior Work:**

The related work is clearly discussed.

**Summary And Contributions:**

The paper introduces a novel benchmark designed to evaluate MLLMs on their ability to understand and reason about generated images. JourneyBench includes five diverse tasks. The benchmark utilizes prompt-based generated images to challenge the models beyond conventional benchmarks, aiming to expose models' biases, hallucination tendencies, and fine-grained perception abilities.

---

> ### Author Rebuttal · Authors · 2024-08-19
>
> ## Total Number of Samples are Insufficient
>
> We appreciate the reviewer’s concern regarding the total number of samples in JourneyBench. We would like to clarify that JourneyBench was designed with a focus on diversity and depth, ensuring that each task within the benchmark challenges models in unique and meaningful ways.
>
> JourneyBench contains 13,631 unique image-text samples across five carefully selected tasks, which include 12,405 unique images and 13,664 unique texts. The benchmark is structured as follows:
>
> - **Complementary Multimodal Chain-of-Thought:** This task includes 2,600 image-question pairs across 10 fine-grained types, specifically categorized based on visual contexts and the need for multimodal co-referencing. This ensures that the task rigorously evaluates models' reasoning capabilities in complex scenarios.
> - **Image Captioning:** The dataset comprises 1,000 images paired with 5,000 captions, where each image is described by five different captions. This task challenges models to generate diverse and accurate descriptions of unconventional and fictional images.
> - **Visual Question Answering (VQA):** JourneyBench includes 7,748 image-question pairs categorized into three types of hallucination triggers. This task is crucial for testing how well models can handle scenarios that are specifically designed to induce hallucinations or erroneous assumptions.
> - **Multi-Image VQA:** With 316 image-question pairs across three fine-grained categories, this subset is larger than the recent multi-image VQA benchmark, Mantis, which contains 217 samples. Our multi-image VQA task is designed to test models' ability to handle intricate visual relationships across multiple images.
> - **Fine-Grained Cross-Modal Retrieval:** This task includes two subtasks:
> - **Image-to-Text Retrieval:** With 1,000 query images paired with 11,121 texts, each image is associated with an average of five positive (ground-truth) texts and six negative (sample-specific distractor) texts.
> - **Text-to-Image Retrieval:** This subtask includes 1,000 samples, each with five ground-truth captions, leading to approximately 5,000 query texts tested against 6,323 images. Each text has one matching image and five negative image distractors.
>
> We designed JourneyBench to ensure that each task not only provides a broad spectrum of challenges but also pushes the boundaries of current vision-language models by introducing tasks that go beyond conventional datasets. While covering every possible edge case in vision-language understanding is a monumental task, JourneyBench focuses on the most critical scenarios that are likely to reveal key weaknesses and biases in models.
>
> We acknowledge that expanding the dataset could further enhance its robustness. Therefore, we are actively working on generating additional samples for future iterations of JourneyBench, aiming to cover an even wider range of scenarios and edge cases. However, the current size and structure of JourneyBench already provide a significant and rigorous evaluation platform that surpasses several existing benchmarks in both size and diversity.

---

> > ### Comment · Reviewer_njtE · 2024-08-20
> > **Official Comment of Reviewer njtE**
> >
> > Thank you for your rebuttal. I will keep my rating.

---

> > > ### Author Response · Authors · 2024-08-24
> > >
> > > Thank you very much for your time, effort, and valuable insights into our work. Your feedback is greatly appreciated and will help us enhance the quality of our research.

---

### Official Review · Reviewer_RFCq · 2024-07-16
**JourneyBench: A Challenging One-Stop Vision-Language Understanding Benchmark of Generated Images**

**Rating:** 7
**Confidence:** 3
**Correctness:** Yes
**Clarity:** Yes

**Review:**

JourneyBench represents a significant advancement in the field of VLU benchmarking, offering a comprehensive suite of tasks that assess the fine-grained multimodal reasoning abilities of MLLMs. The authors have crafted a challenging dataset that pushes the models to their limits, particularly in scenarios involving imaginary and unusual images. The tasks included in JourneyBench are well-designed to probe specific biases and limitations of current models, such as the tendency to hallucinate and the difficulty in understanding fine-grained details.

**Strengths:**

•	Novelty and Relevance: JourneyBench introduces a fresh perspective on VLU benchmarking by focusing on generated images that require models to reason about unconventional scenarios. This is a timely contribution given the increasing prevalence of generated imagery in online media.
•	Comprehensive Assessment: The five tasks encompass a wide range of reasoning abilities, providing a thorough examination of models' performance in diverse conditions.
•	Controlled Experimentation: The use of generated images allows for controlled experimentation, enabling the assessment of models in scenarios that can be precisely tailored to test specific reasoning capabilities.

**Additional Feedback:**

Diversity of Scenarios: While the paper highlights the diversity of generated images, a deeper analysis of the breadth of scenarios covered could be provided to ensure that the benchmark truly tests the limits of models across a wide spectrum of reasoning tasks.

**Documentation:**

No

**Limitations:**

Yes

**Opportunities For Improvement:**

Diversity of Scenarios: While the paper highlights the diversity of generated images, a deeper analysis of the breadth of scenarios covered could be provided to ensure that the benchmark truly tests the limits of models across a wide spectrum of reasoning tasks.

**Relation To Prior Work:**

Yes

**Summary And Contributions:**

JourneyBench aims to fill a gap in the evaluation of MLLMs by introducing a diverse set of tasks that require models to demonstrate advanced reasoning abilities in atypical visual contexts. The benchmark emphasizes the importance of testing models in scenarios where language bias and holistic image gist are insufficient, thereby promoting a deeper understanding of models' visual reasoning capabilities. The paper discusses the methodology behind the creation of the benchmark, including the use of a human-machine-in-the-loop (HMIL) framework to generate and curate the images and questions.

---

> ### Author Rebuttal · Authors · 2024-08-19
>
> ## Diversity of Scenarios
>
> We agree that further exploration of the diversity within the dataset could strengthen the paper. We provide detailed 11 categories for image captioning and cross-modal retrieval, 10 categories for chain-of-thought, 5 categories for visual question answering, and 3 categories for multi-image reasoning. The detailed analysis can be found in Figure 4 in the main paper, Table 3, Table 4, Table 6, Figure 6, Figure 7, Figure 8, Figure 9, and Figure 10 in the Appendix. We will include additional analyses in the revised version to better illustrate the range of scenarios included in JourneyBench, highlighting how these scenarios push the limits of current models' reasoning abilities.

---

### Official Review · Reviewer_uNZR · 2024-07-20
**Review for JourneyBench paper**

**Rating:** 6
**Confidence:** 5
**Clarity:** Yes

**Review:**

This paper presents a new benchmark for evaluating VLMs in various tasks, including image-text retrieval, arithmetic VQA, imaginary image captioning, and VQA with hallucinations or invalid assumptions. It features a large, test-only dataset of image-text pairs designed to fill gaps in previous VLM evaluations. Key contributions include its size, new dimensions for evaluation, a robust VQA protocol, and detailed documentation that helps in understanding of VLM weaknesses. Strengths of the paper include its comprehensive dataset and innovative evaluation methods. However, weaknesses include small subset sizes for certain tasks, incomplete comparisons with human preference learning models, lack of detailed error analysis, and ethical concerns regarding the representation of real people in generated images.
Read Strengths and Opportunities For Improvement for more details.

**Strengths:**

The paper presents a comprehensive, large, and well-documented test dataset designed to evaluate VLMs across several understudied tasks, which is a significant contribution to the field.

The paper introduces several new dimensions for evaluating VLMs, including 2-image reasoning and hallucination, two major problems in current models. It also tests several state-of-the-art models.

The dataset breaks down the methods' evaluation into fine-grained categories, which helps to determine which models perform best on specific tasks.

The evaluation protocol for VQA tasks used in the paper is better than previous string matching or multiple choice loss calculation methods. While not a direct contribution, using best practices is a strength for any paper.

**Additional Feedback:**

NA

**Correctness:**

Overall, the claims made in the submission appear to be largely accurate and the data set is well constructed. However, there are some concerns about the relevance of the results due to some subset sizes, missing model comparisons, detailed failure analysis, and ethical acknowledgements.

**Documentation:**

Overall, the data is well documented. The dataset is well appreciated, but certain questions need to be better addressed.
-Does the dataset relate to people?
-Does the dataset identify any subpopulations (e.g.,
by age, gender)?
-Is it possible to identify individuals (i.e., one or more
natural persons), either directly or indirectly (i.e., in
combination with other data) from the dataset?

**Ethics:**

Ethical concerns regarding the depiction of real people in generated images. It is important to address these concerns by acknowledging the representation of real people and considering the potential implications. The dataset documentation claims that it is not possible to identify individuals, arguing that the dataset is completely generated. However, this is questionable. For example, an image labeled "aktiediplomaten_Donald_Trump_and_Vladimir_Putin_drinking_champa_39af67bf-37ba-4870-89d1-4a5e3dd9ced1.png" clearly identifies real people and adds country references. The corresponding question also explicitly refers to these individuals. While the use of such data is not necessarily problematic, it is crucial to properly acknowledge the content and implications of including such information, especially when it comes to politically connected figures. The community needs to do a better job of defining and adhering to privacy policies when using generated images.

**Limitations:**

The limitations of the work are discussed in the supplementary material, addressing several points. Two points that should be better handled are: 1) the fact that the tasks within the dataset are described as "extremely challenging," raising the question of whether they are solvable at all. A baseline model should provide some intuition about this (refer to weaknesses); and 2) the ethical and privacy concerns regarding the use of generated images containing humans.

**Opportunities For Improvement:**

1. The small size of some task subsets, such as the multi-image VQA, may limit the significance of the results. Specifically, there are only 316 samples for the multi-image VQA across three categories. With this limited number of samples, the significance of the multi-image VQA evaluation is questionable. For a more comprehensive study, it would be better to include all tasks with a significant number of test samples. One solution is to increase the sample size for smaller subsets.

2. Incomplete comparison with human preference-learned models. The comparison of pre-trained models is missing a whole range of models, in particular the human preference-learned models that are relevant for the type of dataset proposed in this work. Models such as ImgeReward [1], Pick-a-pic [2], HPS [3], and VQAscore [4] are notably absent. In addition, most of these models are trained on generated images, which may help to understand the significant gap between models trained on real images and optimal performance. A possible solution is to include comparisons with human preference-learned models to provide a more comprehensive evaluation.

3. Lack of detailed insight into specific areas where VLMs fail. After reading the paper, I expected to gain a clearer understanding of where VLMs fail. For example, in the retrieval task, there seems to be no significant differences between the models, while in the imaginary image captioning task, some models outperform others. A major difference between these models is their training and the datasets used for training, especially considering the large gap between using real and generated images for evaluation, and the fact that these models were trained using a specific language and question formulation.

To improve understanding, I suggest two things:
a. Compile a more detailed analysis of VLM errors and provide a guide for improvements.
b. Train a baseline model (from a pre-trained model) using these guidelines to show that it outperforms the pre-trained version on the identified weaknesses. This baseline can also show whether the dataset itself is feasible for models by training with specific goals, or simply encoded in a hard procedure for actual VLMs.

4. Ethical concerns regarding the depiction of real people in generated images. It is important to address these concerns by acknowledging the representation of real people and considering the potential implications. The dataset documentation claims that it is not possible to identify individuals, arguing that the dataset is completely generated. However, this is questionable. For example, an image labeled "aktiediplomaten_Donald_Trump_and_Vladimir_Putin_drinking_champa_39af67bf-37ba-4870-89d1-4a5e3dd9ced1.png" clearly identifies real people and adds country references. The corresponding question also explicitly refers to these individuals. While the use of such data is not necessarily problematic, it is crucial to properly acknowledge the content and implications of including such information, especially when it comes to politically connected figures. The community needs to do a better job of defining and adhering to privacy policies when using generated images.

[1] Xu, Jiazheng, et al. "Imagereward: Learning and evaluating human preferences for text-to-image generation." Advances in Neural Information Processing Systems 36 (2024).
[2] Kirstain, Yuval, et al. "Pick-a-pic: An open dataset of user preferences for text-to-image generation." Advances in Neural Information Processing Systems 36 (2023): 36652-36663.
[3] Wu, Xiaoshi, et al. "Human preference score: Better aligning text-to-image models with human preference." Proceedings of the IEEE/CVF International Conference on Computer Vision. 2023.
[4] Li, Baiqi, et al. "Evaluating and Improving Compositional Text-to-Visual Generation." Proceedings of the IEEE/CVF Conference on Computer Vision and Pattern Recognition. 2024.

**Relation To Prior Work:**

Refer to weaknesses

**Summary And Contributions:**

The paper proposes a new benchmark for evaluating Vision-Language Models (VLMs) in various tasks, specifically:

    1. Image-Text Retrieval
    2. Cross-reference arithmetic VQA
    3. Multi-image arithmetic VQA and A vs. B VQA
    4. Imaginary image captioning
    5. VQA with Hallucination or Invalid Assumptions

It introduces a new test-only dataset containing 13,631 unique image-text pairs, including 12,405 unique images and 13,664 unique text instances. This dataset aims to address gaps in VLM evaluations not explicitly covered by previous work.

Contributions:
1. The dataset is large for a test set aimed at evaluating VLMs.
2. The work addresses several new dimensions of VLM evaluation.
3. The evaluation protocol uses LLM verification for VQA, a more robust approach than exact text matching or other previous VQA evaluations.
4. The dataset provides a fine-grained breakdown of performance, which helps to identify specific weaknesses in VLMs.
5. The paper includes comprehensive documentation of the dataset, detailing the number of points per item, item types, and the procedures used to construct the dataset.

---

> ### Author Rebuttal · Authors · 2024-08-19
>
> ## 1. Task Subset Size
> We acknowledge the concern regarding the subset sizes, particularly for the multi-image VQA task. We chose a limited number of samples for this task due to the novelty and difficulty in generating high-quality data in this domain, especially for multi-image VQA, which requires two related images for one single data point. Despite the challenging nature of this task, we managed to collect 316 image-question samples for our multi-image VQA task, which is still larger than existing multi-image VQA benchmark, Mantis-Eval (217 samples). To enhance the robustness of our benchmark, we are in the process of expanding this subset by generating additional samples, which will be included in a future version of the dataset. We emphasize, however, that our current dataset size for the multi-image VQA task, while small, is still larger than existing work.
>
> We would also like to emphasize that JourneyBench, as a whole, is a comprehensive benchmark comprising 13,631 unique image-text samples across five tasks. This ensures broad coverage and robustness. For instance, JourneyBench includes 2,600 image-question pairs for complementary multimodal chain-of-thought, categorized into 10 fine-grained types based on visual contexts and multimodal co-referencing. The image captioning dataset contains 1,000 images paired with 5,000 captions, while the visual question answering task comprises 7,748 image questions, categorized into three fine-grained types of hallucination triggers. Additionally, our fine-grained cross-modal retrieval task features two subtasks, each with thousands of carefully annotated samples.
>
> Given this extensive scope, we believe that the significance of our results is well-supported by the breadth and depth of the overall dataset, even as we continue to enhance specific subsets such as multi-image VQA.
>
> ## 2. Comparison with Human Preference-Learned Models
>
> We appreciate this suggestion and agree that including these models would provide a more comprehensive evaluation. We have evaluated Pick-a-Pic on our retrieval dataset and reported the result in the PDF. We will include comparisons with more human preference-learned models, e.g. ImageReward, HPS, and VQAscore, in our next round of experiments. This comparison will be highlighted in the final version of the paper to provide deeper insights into model performance.
>
> ## 3. Detailed Error Analysis and Baseline Model
>
> We appreciate the reviewer’s feedback and agree that providing a more detailed analysis of VLMs' failure cases would offer valuable insights into model performance. Our benchmark is designed with fine-grained categorization to facilitate this type of analysis. Specifically:
>
> - **Image Captioning and Cross-Modal Retrieval:** We categorized the tasks into 11 detailed categories, such as unusual construction and strange scenes. As illustrated in Figure 4 of the main paper, existing models clearly struggle with these more complex visual scenarios, revealing significant gaps in their capabilities.
>
> - **Chain-of-Thought Reasoning:** This task is divided into 10 categories, including challenges like hallucination, consistency, and handling large numbers. Figure 5 shows that while models generally struggle with these scenarios, proprietary models tend to perform better in tasks involving large numbers. This suggests a disparity in how different models handle numerical reasoning.
>
> - **Visual Question Answering:** The VQA task (based on HaloQuest) includes 3 categories of different hallucination triggers and 2 categories of samples with generated or real images.
>
> - **Multi-Image Reasoning:** This task includes 3 categories, and our analysis shows that models excel in cause-and-effect reasoning but struggle with arithmetic reasoning and tasks requiring external knowledge, as highlighted in Table 3. These differences are particularly pronounced when comparing open-source models to proprietary ones.
>
> Due to space limitations in the main paper, we included additional detailed analyses in the Appendix (Table 3, Table 4, Table 6, Figure 6, Figure 7, Figure 8, Figure 9, and Figure 10). These sections offer a more granular look at model performance, providing quantitative results that highlight specific areas where models fail and excel.
>
> In response to your suggestion, we will further enhance the paper by adding a more detailed error analysis. For instance, in the multimodal chain-of-thought reasoning task, we plan to explore different types of hallucination errors more deeply—examining whether models are miscounting numerical values or if they are hallucinating about the language context itself.
>
> Regarding the baseline model, we are considering training a baseline specifically tailored to address the identified weaknesses in our benchmark. This model would be fine-tuned on our dataset with targeted objectives, such as improving performance in categories where models currently struggle. By comparing this baseline to pre-trained versions, we aim to demonstrate the feasibility and effectiveness of our dataset in improving VLM performance.
>
> ## 4. Ethical Concerns Regarding Real People in Generated Images
>
> The dataset was created with the intent to avoid directly identifiable individuals, and certain generated images may resemble real-world figures of celebrities. To address this, we will implement a systematic approach to detect named entities within the dataset to ensure that only public figures are referenced. Additionally, we will include a clear disclaimer in the dataset documentation, explicitly stating that these images are AI-generated and not real, to ensure that downstream users are fully aware of the artificial nature of the content. To the best of our knowledge, our dataset does not include any requests to generate images of specific non-celebrity individuals. Nonetheless, we will thoroughly review the dataset to confirm this and will update the documentation to acknowledge any potential ethical implications.

---

> > ### Comment · Reviewer_uNZR · 2024-08-23
> > **Response to rebuttal**
> >
> > I appreciate the authors' response.
> >
> > Regarding the discussion on error gaps and the statement that "existing models clearly struggle with these more complex visual scenarios," this increases the concerns about the feasibility of the tasks presented. To ensure that these tasks are solvable, it is strongly recommended to include a baseline model in the benchmark, even in an ideal/oracle scenario. This would provide a clearer indication of the challenges involved in solving the tasks and clarify their feasibility.
> >
> > Regarding the detailed error analysis, Figures 6, 7, 8, 9, and 10 in the supplementary present statistics related to the dataset rather than model performance. While these statistics can help in interpreting results, they do not directly address my earlier question. It's important to include an analysis that directly links the model's training data, performance, and the evaluation dataset, particularly in terms of data size and relevance. As Reviewer RFCq also noted, a deeper analysis across a broader range of scenarios is necessary. Presenting results without relevant dataset statistics, or statistics without relevant results and training data, leaves an incomplete picture. Shuch analysis could significantly contribute to understanding which techniques are most effective and where models might fall short.
> >
> > **Comparison with Human Preference-Learned Models**
> > Thank you for including the Pick-a-Pic model. However, I noticed that no analysis was provided regarding the results or the inclusion of other methods. Additionally, I find it surprising that the CLIP model outperforms others in this context. A discussion on these results would help clarify these outcomes.
> >
> > **Additional Comments:**
> > - Regarding the expansion of the dataset for the multi-image VQA task, could you clarify the size of the data you plan to release?
> > - I observed that Table 3 in the supplementary material is almost identical to Table 3 in the main paper, with the exception of one row. It would be more efficient to consolidate this information into a single table in the main paper.
> > - When releasing a dataset involving human content, it's crucial to apply rigorous review and filtering processes. This includes blurring faces and filtering for NSFW content, especially when the intent is to focus on the overall context rather than individual human subjects.
> >
> > After reviewing the rebuttal, I maintain my original score.

---

### Author Rebuttal · Authors · 2024-08-19

We would like to express our sincere appreciation to the reviewers for their thorough and constructive feedback. We appreciate the reviewers' recognition of the novelty and impact of JourneyBench in advancing Vision-Language Model (VLM) evaluation, particularly through our diverse and challenging tasks with high-quality data using generated and real images. Reviewers consistently praised the comprehensiveness of our dataset, the robustness of our evaluation methods, and the detailed analysis of existing models in fine-grained dimensions.

The primary concerns raised include the limited sample size in some tasks, the need for comparisons with human preference-learned models, more detailed error analysis, and ethical considerations related to the representation of real people in generated images. We have carefully considered all the comments and suggestions, and we provide the following clarifications and proposed revisions.

We hope these revisions address the reviewers' concerns and demonstrate our commitment to improving the quality and impact of JourneyBench. We are confident that these changes will enhance the contribution of our work to the field of Vision-Language Understanding.

Thank you once again for your valuable feedback!

---

### Decision · Program_Chairs · 2024-09-26

**Decision:**

Accept (Poster)

**Comment:**

JourneyBench aims to use generated images and a Human-machine-in-loop interface for annotating these images. This allows the authors to build a dataset that is completely generated and contains challenging images which require attention to fine details by the model to perform well.

Strengths of this paper include the use of LLM + VLMs + human filtering / checking. The use of AI based automated tooling is an important next step for producing challenging datasets for the next generation of models. THis is particularly important for multi-modal datasets, which are challenging to capture and annotate solely by hand. They cover a wide variety of scenarios with this dataset and present a clever way of iteratively annotating the data. They also present a wide variety of experiments demonstrating that this benchmark is more challenging than other benchmarks (e.g. MS-COCO).

A common weakness identified by multiple reviewers is that the sample size is relatively small for some of the tasks, particularly the multi-image VQA. The authors point out that this is comparable to one existing multi-VQA benchmark, but even for other tasks, there is typically only ~1000 samples, which is relatively low. There are also some minor issues with the presentation (much of the dataset statistics are in the supplementary material due to space and a lack of error analysis / insight ... what kinds of situations are these models failing on frequently / what fine-grained insights can we gain about building training sets?).

One reviewer mentions that a deeper analysis of the failures would be useful. This is partially addressed in the rebuttal (a substantive categorization of the errors by question category is provided), but there is still little to go on that explains why particular models may struggle beyond complex visual data seems to cause problems.

Overall I recommend acceptance, while there are some concerns with the number of samples in the benchmark and a lack of analysis on why particular models do better over others, I think the benefits of this benchmark outweigh the issues. The authors should consider strengthening their paper with additional discussion of the types of errors uncovered with this dataset.